# Isolation of *Bacillus siamensis* B-612, a Strain That Is Resistant to Rice Blast Disease and an Investigation of the Mechanisms Responsible for Suppressing Rice Blast Fungus

**DOI:** 10.3390/ijms24108513

**Published:** 2023-05-10

**Authors:** Yanmei Yang, Yifan Zhang, Luyi Zhang, Zhanmei Zhou, Jia Zhang, Jinchang Yang, Xiaoling Gao, Rongjun Chen, Zhengjian Huang, Zhengjun Xu, Lihua Li

**Affiliations:** 1Crop Ecophysiolgy and Cultivation Key Laboratory of Sichuan Province, Sichuan Agricultural University, Chengdu 611130, China; yym1578073713@163.com (Y.Y.); lacrimosez@163.com (Y.Z.); zhangluyi060@gmail.com (L.Z.); zmfeb1996@163.com (Z.Z.); zjbio96@163.com (J.Z.); amyriceworld@hotmail.com (X.G.); chenrj8@aliyun.com (R.C.); phosphate@126.com (Z.H.); 2Maize Research Institute of Sichuan Agricultural University, Chengdu 611130, China; yjc99726@163.com

**Keywords:** *Bacillus siamensis*, rice blast, defense genes, conidial germination

## Abstract

Rice yield can be significantly impacted by rice blast disease. In this investigation, an endophytic strain of *Bacillus siamensis* that exhibited a potent inhibitory effect on the growth of rice blast was isolated from healthy cauliflower leaves. 16S rDNA gene sequence analysis showed that it belongs to the genus *Bacillus siamensis*. Using the rice *OsActin* gene as an internal control, we analyzed the expression levels of genes related to the defense response of rice. Analysis showed that the expression levels of genes related to the defense response in rice were significantly upregulated 48 h after treatment. In addition, peroxidase (POD) activity gradually increased after treatment with B-612 fermentation solution and peaked 48 h after inoculation. These findings clearly demonstrated that the 1-butanol crude extract of B-612 retarded and inhibited conidial germination as well as the development of appressorium. The results of field experiments showed that treatment with B-612 fermentation solution and B-612 bacterial solution significantly reduced the severity of the disease before the seedling stage of Lijiangxintuan (LTH) was infected with rice blast. Future studies will focus on exploring whether *Bacillus siamensis* B-612 produces new lipopeptides and will apply proteomic and transcriptomic approaches to investigate the signaling pathways involved in its antimicrobial effects.

## 1. Introduction

Rice (*Oryza sativa* L.) is one of the most significant food crops in the world and provides 21% of all calories consumed globally [1]. Rice blast, often known as rice cancer, is the most dangerous disease that can affect rice on a global level. Thus far, rice blast disease has been reported in 85 nations, with the worst incidence occurring in Asia and Africa [2]. *Magnaporthe oryzae* (*M. oryzae*) is a filamentous ascomycete fungus and is ranked as one of the ten most important plant pathogens; this fungus is spread aerially by conidia during an epidemic [3].

There are several methods used to combat rice blast disease at present, including the selection and breeding of disease-resistant types, chemical control, and biological control. Although infections prefer adaption to overcome host plant resistance, the formation of resistant types takes significant time; furthermore, resistance is not universal nor permanent. As a result, the breakdown of rice resistance is caused by the quick development of new strains. There are several long-term consequences associated with the careless use of chemical fertilizers and pesticides, including environmental damage, higher production costs, and the leaching of chemicals into food [4]. Biological control is a method used to eradicate and suppress pests that make use of other species with beneficial characteristics [5].

Endophytes are symbiotic microorganisms, primarily bacterial and fungal species, that live in seemingly healthy internal plant tissues without harming the host plants [6]. Due to the fact that they are spore-forming bacteria, beneficial strains of *Bacillus* have significant potential as biocontrol agents because they are simple to cultivate and preserve as inoculants [7]. In a previous paper, Rong et al. isolated an endophytic strain (*Bacillus safensis* B21) from cinnamon fruits; the secondary metabolites iturin A2 and iturin A6 isolated from this strain were found to inhibit the growth of *M. oryzae* mycelium by altering the permeability of the mycelial membrane [8]. In another study, Huang et al. obtained a sample of *Bacillus* cereus (HS24) from a rice farm; at a concentration of 10^7^ colony-forming units (CFU)/mL, this bacterium effectively suppressed *M. oryzae* conidial germination by 97.8% [7].

There are, however, limited reports relating to the ability of *Bacillus siamensis* to inhibit rice blast, thus necessitating additional research into the potential mechanisms by which this strain can exert influence on rice blast. In this study, an endophytic bacterium with strong growth-inhibitory properties was isolated from healthy cauliflower leaves and given the designation B-612. This bacterium was identified by morphology and 16S rDNA sequence analysis. Then, we investigated the susceptibility of B-612 to 20 different antibiotics and how each of these antibiotics affected mycelial development, attachment formation, and conidial germination. Quantitative real-time polymerase chain reaction (qRT-PCR) was used to investigate the expression of genes related to the defense mechanism of rice and hydrogen peroxide (H_2_O_2_) buildup following treatment with B-612 fermentation broth. Field tests were also carried out to support the potential of B-612 fermentation as a biological defense mechanism against rice blast.

## 2. Results

### 2.1. Isolation and Identification of Endophytes

As shown in Figure 1a, B-612 showed a strong inhibitory effect against *M. oryzae* Guy11. The endophytic bacterium B-612, which was isolated from cauliflower leaves, inhibited the growth of *M. oryzae* by up to 79.76% according to tests for mycelial growth inhibition. When inoculated with ring picking bacteria, we found that endophyte B-612 could develop normally on beef paste peptone plate, and that the colonies were round, white, and opaque. The surface of the colonies was flat, dry, and wrinkled, and the interior was mucilaginous (Figure 1b). *Bacillus siamensis* KCTC 13613 and strain B-612 were found to be almost identical by 16S rRNA sequencing analysis. Figure 1c shows a phylogenetic tree illustrating the link between strain B-612 and other strains. Thus, strain B-612 was identified as *Bacillus siamensis*.

### 2.2. Sensitivity of Strain B-612 to Various Antibiotics

As shown in Figure 2, strain B-612 grew normally on plates containing erythromycin (15 μg) but formed clear circles on plates containing other antibiotics. *Siamensis* B-612 was susceptible to most of the antibiotics tested, including penicillin (10 μg), qxacillin (30 μg), ampicillin (10 μg), carbenicillin (100 μg), piperacillin (100 μg), cephalexin (30 μg), cefamezin(30 μg), cefradine (30 μg), cefuroxim (30 μg), ceftazidime (30 μg), ceftriaxone (30 μg), cefoperazone (75 μg), doxycycline (30 μg), amikacin (30 μg), gentamicin (10 μg), kanamycin (30 μg), neomycin (30 μg), tetracycline (30 μg), and minocycline (30 μg). The frequent use of antibiotic-resistant bacteria can be harmful to the environment, while *Siamensis* B-612 used in this study is sensitive to most antibiotics and has the potential to be used as a biocontrol bacterium [9,10].

### 2.3. B-612 Fermentation Solution Enhanced the Resistance of Rice to M. oryzae

In order to detect the expression of peroxidase (POD) and genes related to the defense response in rice, we used rice *OsActin* genes as internal references. These genes fell into three main categories: first, genes related to disease-course, such as the *OsPR1a*, *OsPR5a*, and *OsPR10a* families; second, genes related to rice signal transduction, such as the salicylic acid (SA) signaling receptor gene *OsNH1*, chitin signaling receptor genes *OsCEBiP* and *OsLYP6*, and mitogen-activated protein kinase (MAPK) pathway genes; and third, genes encoding transcription factors, including *OsWRKY45*, *OsWRKY89*, *OsWRKY53*, and *OsEREBP* [11,12,13,14,15,16]. Between 24 and 48 h after inoculation, rice defense genes were activated to varying degrees (Figure 3). All of these genes showed a significant upregulation at 48 h after treatment, with the expression levels of *OsCEBiP* and *OsNH1* peaking at 24 h and the remaining genes peaking at 48 h. Comparing the B-612 fermentation broth-treated group to the control group at 48 h, *OsPR1a* expression was upregulated by approximately 60-fold, *OsPR5* expression by approximately 150-fold, and *OsLYP6* expression by approximately 40-fold. POD activity increased gradually and peaked at 48 h after inoculation, exhibiting an approximately 80-fold rise in POD levels in comparison to 0 h. After 48 h, it was evident that the group treated with B-612 fermentation broth had POD expression levels that had been upregulated by approximately 20-fold when compared to the control group. These data showed that the significant biocontrol action of B-612 was caused by the increased expression of defense-related genes and that this plays a crucial role in the defense of rice against rice blast.

### 2.4. H_2_O_2_ Accumulation

The main function of superoxide dismutases (SODs) is to scavenge cellular superoxide radicals (O^2−^). However, the production of SOD can provide additional protection against pathogenic infections in plants, and increased SOD activity has previously been shown to lead to the accumulation of H_2_O_2_ [17,18]. In this study, we used 3,3′-diaminobenzidine tetrahydrochloride (DAB) staining to detect H_2_O_2_ accumulations, as depicted in Figure 4. The concentration of H_2_O_2_ in the rice leaves treated with fermentation broth increased over time, and peaked at 48 h.

### 2.5. Detection of Lipopeptide Biosynthetic Genes

Our PCR analysis indicated that strain B-612 expressed many relevant and functional genes, including fengycin (*fenB*, *fenD*), iturin (*ituD*), surfactin (*srfAA*), bacillomycin (*bmyB*), and biotene (*bioA*). These genes were amplified using primer pairs and generated PCR products with the predicted sizes of 1400, 293, 1203, 273, 395, and 210 base pairs (Figure 5). BLASTX analysis revealed a 96.26% amino acid sequence similarity between strain B-612 and *Bacillus subtilis* with regards to bacillin synthase, thus indicating that B-612 is able to successfully generate bacillomycin synthase and bacillomycin. The amino acid sequence similarity between strain B-612 and fengycin synthase B of *Bacillus amyloliquefaciens* and *Bacillus velezensis* was 100%, while the amino acid sequence similarity between fengycin synthase D of *Bacillus subtilis* was 98.68%. This shows that B-612 is able to synthesize fengycin synthetase B and fungogenin synth. Strain B-612 and *Bacillus subtilis* share 100% of amino acid identified for surfactin synthase; this suggests that B-612 may generate surfactin synthase on its own. *Bacillus* sp. and strain B-612 shared an amino acid sequence that was more than 98% similar with regards to iturin synthase, thus showing that B-612 can synthesize iturin synthase to make iturin. Thus, strain B-612 has the potential to prevent the growth of *M. oryzae* by producing key lipopeptide antibiotics, including fengycin, iturin, surfactin, or bacillomycin.

### 2.6. The Effects of Antifungal Substances Generated by Strain B-612 on M. oryzae Conidia Germination and the Formation of Appressorium

*M. oryzae* conidia were treated with either distilled water or a crude extract of B-612.

The conidia in the control group began to develop budding tubes after two hours, as shown in Figure 6. The creation rate was 72.33% when the budding tubes extended and connected cells formed at the other end after 8 h. After being exposed to B-612 crude extract, the spores had hardly begun to develop budding tubes at 2 h; at 8 h, very few of the spores had generated budding tubes. The budding tube germination rate remained below 10% during the observation period of 48 h, and breakage of the connected cells was evident at both 24 and 48 h. Throughout the experiment, the development of normally linked cells was not apparent. At both 24 and 48 h, it was evident that the budding tubes had broken, and that generally, the normal development of adherent cells was not apparent.

### 2.7. Biocontrol Efficacy of Strain B-612 and Its Culture Filtrate

Field tests were carried out to assess the ability of strain B-612 to prevent rice blast. Before being exposed to rice blast at the seedling stage in Lijiangxintuan (LTH), the treatment groups were administered B-612 filter fermentation solution and B-612 bacterial solution treatments; the administration of water and LB medium treatments served as positive controls while the administration of carbendazim treatments served as negative controls. Analysis revealed large spots in the positive control group (water and LB) with yellowish edges and dark brown centers and obvious signs of dieback; the spots in the positive control group (carbendazim) were small and less susceptible while the spots in the B-612 filtrate fermentation solution and B-612 mycorrhizal solution treatment groups were brown in color without signs of dieback. The group treated with fermentation solution was less susceptible than the other two groups. Counting the number of spots within 5 cm of the leaf length, we found that there were significantly fewer spots in the treated group than in the negative control group (Figure 7) (*p* < 0.01). Thus, it appears that B-612 can be employed as a biocontrol bacterium for the control of rice blast in the field and that its mycorrhizal and filter fermentation solutions are more efficient in preventing rice blast on rice leaves.

## 3. Discussion

According to previous research, endophytic bacteria directly produce bioactive secondary metabolites that help protect their host plants from pathogenic microbes, thus improving the fitness of their hosts [19]. One benefit of a genus for biological control is its capacity to produce spores and endure harsh environments. Endophytic *Bacillus* species offer a variety of advantages to plants, including defense against insects, nematodes, and pathogenic microbes; they can also induce resistance and foster plant growth without harming the environment [20]. *Bacillus siamensis* has been reported to reduce the incidence of a variety of plant diseases, including tobacco brown spot, rice leaf blight, and spike blight [21,22,23,24]. In a previous study, Xu et al. purified iturin A and bacillomycin F from *Bacillus siamensis* JFL15 to inhibit the growth of *M. oryzae* but did not investigate the precise mechanism involved. In the present study, a *Bacillus siamensis* B-612 endophytic strain was isolated from healthy cauliflower leaves and showed a potent inhibitory effect on the growth of rice fungus. However, it was also responsive to various antibiotics. Thus, *Bacillus siamensis* B-612 has the potential to be employed as a biocontrol bacterium to control rice disease.

Airborne conidia generated by *M. oryzae* play a significant role in the spread and severity of rice blast. The conidia germinate after touching a suitable host surface, thus creating an attachment cell, a dome-shaped infection structure at the end of the germ tube [25,26,27]. A variety of *Bacillus* species have been shown to inhibit the conidial germination of *M. oryzae* [28,29]. Yet, the conidia germination test findings performed in our present study showed that the germination process had been evidently reduced or even postponed. These findings clearly demonstrated that the 1-butanol crude extract of B-612 retarded and inhibited conidial germination as well as the development of appressorium. As a result, we can infer that the crude 1-butanol extract of B-612 can prevent *M. oryzae* from forming infectious structures in vitro. This is the first study showing that *Bacillus siamensis* can inhibit the spore germination and cell attachment formation of *M. oryzae*.

The biocontrol ability of *Bacillus* strains and their capability for global adaptation in their natural habitat, including settlement and biofilm development, depend on efficient lipopeptide production. Among plant-associated Bacillus isolates, the biocontrol agents (BCA) genes bmyB, srfAA, and fenD are most frequently observed [30]. It has been reported that when distinct lipopeptide families are synthesized together, their interactions can become synergistic and improve the activity of each individual family member [31,32,33]. Experimental results showed that B-612 may suppress *M. oryzae* growth by secreting antimicrobial lipopeptides such as bacillomycin, fengycin, iturin, or surface activator; however, these compounds still need to be further identified and purified.

The production of antibiotics, competition, the promotion of plant development, and the induction of systemic acquired resistance (SAR) and induced systemic resistance (ISR) are among the primary processes of BCA [30]. Following the production of SA, the coordinated activation of several PR proteins, and resistance-associated enzymes, SAR can initiate defense responses in response to the detection of pathogenic invasion [34]. BCA-induced SOD synthesis can provide plants with enhanced defense against harmful diseases [18]. Pathogen-sensing signaling molecules, MAPK, and transcription factors all contribute to the defense response of plants [13]. In a previous study, Awan et al. reported that *Bacillus siamensis* reduced cadmium toxicity in wheat plants by enhancing the antioxidant defense system [35]. In another study, Zhou et al. reported that *Bacillus siamensis* YC-9 increased the activities of defense enzymes such as POD, polyphenol oxidase (PPO), and phenylalanine aminolase (PAL) in the roots of diseased cucumber, thus resulting in increased levels of resistance [36]. In the present study, we showed, for the first time, that *Bacillus siamensis* can enhance systemic resistance in rice by inducing the expression of defense genes. These findings demonstrated that one of the primary mechanisms of action of B-612 fermentation broth was to induce SAR; more specifically, when B-612 fermentation broth was administered, free SA was created and the expression levels of SA-dependent PR family genes were markedly elevated. B-612 fermentation broth significantly increased the expression of rice defense genes 48 h after treatment, thus indicating that these genes and the products they produce may be important in the control of resistance during the late stages of *M. oryzae* infestation. During treatment, POD activity gradually increased, peaking 48 h after inoculation; these outcomes were in line with H_2_O_2_ buildup. The enzymatic activity of POD contributes to its role in disease resistance and can stop the oxidation of damaged plant tissues. Inhibiting the growth of rice blast and minimizing the oxidative damage produced by rice blast may have been achieved by treating rice with B-612 fermentation solution.

In conclusion, B-612 can activate the defensive system of rice plants, thus suppressing rice blast, although further research into the metabolic pathway of activation is necessary. Field tests were used to further investigate the ability of B-612 to prevent rice blast disease. These findings demonstrated that disease severity was significantly reduced following treatment with B-612 fermentation solution and B-612 bacterial solution at the seedling stage of LTH prior to infection with rice blast disease, and that the effects of this treatment were comparable to carbendazim spraying. In order to effectively prevent rice blast disease in the field, B-612 fermentation solution could be sprayed during the pre-susceptible stage. Collectively, these results suggest that *Bacillus siamensis* B-612 is a promising biocontrol agent for the effective control of rice blast disease. Future studies will focus on exploring whether *Bacillus siamensis* B-612 produces new lipopeptides and will apply proteomic and transcriptomic approaches to investigate the signaling pathways involved in its antimicrobial effects. In the meantime, the B-612 genome will be sequenced to confirm that this is not the previously characterized *Bacillus siamensis* strain.

## 4. Methods

### 4.1. Isolation and Cultivation of Endophytic Bacteria

Fresh, healthy cauliflower leaves were collected, cleaned with distilled water, and placed in sterile 50 mL centrifuge tubes for storage. Then, we sterilized the leaves by submerging them two-to-three times in 75% ethanol for 3 min. Then, the leaves were placed in a solution of 1% sodium hypochlorite and soaked for two minutes; the leaves were then rinsed repeatedly with sterile distilled water to remove sodium hypochlorite from their surfaces. After drying the leaves and storing them in a 10 mL sterile centrifuge tube, a volume of sterile distilled water was added and the mixture was ground until it was homogeneous. Next, we spread 150–200 μL of leaf homogenate on a solid beef paste peptone substrate following the preparation of a gradient of dilutions with sterile distilled water. The dishes were incubated for 24–36 h at 37 °C in a constant temperature incubator. Individual colonies on the plates were chosen, transferred with clean toothpicks into LB liquid medium, and then incubated at 37 °C at 200 rpm for 36–48 h in a constant temperature shaking isolator. The endophytic bacterial solution was provided with the appropriate quantity of glycerol and kept as a backup at −80 °C. The dishes were coated with sterile distilled water for 48 h to prevent non-endophytic bacteria from adhering to the surface of the leaves and thus influencing test outcomes. To ascertain whether all of the non-endophytic bacteria on the surface of the leaves had been eliminated, we monitored the surfaces of the plates for the development of bacteria.

### 4.2. Rice Blast Pathogen (M. oryzae Guy11) and Culture Conditions

The Plant Pathogenic Laboratory of Sichuan Agriculture University provided us with the rice blast pathogenic fungus *M. oryzae* Guy11. This fungus was cultured on potato dextrose agar (PDA) at 28 °C.

### 4.3. Fungal Antagonism Assay

For fungal antagonism assays, 100 mL of LB liquid medium was combined with endophytic bacterial broth that had been kept at −80 °C and cultured for 72 h at 37 °C and 200 rpm in a constant temperature shaker. To create the endophytic filtered bacterial fermentation broth, the fully fermented endophytic bacterial broth was centrifuged at 4 °C at 11,000 r/min for 20 min. The supernatant was then filtered and sterilized with a 0.22 μm filter. Next, the plate was shaken and 5 mL of the filtrate fermentation solution was poured into 100 mL of chilled, uncoagulated PDA medium. To create drug-containing PDA plates, the filtrate fermentation solution-containing PDA plates were chilled and blown dry. A hole punch (7 mm diameter) was used to create a number of *M. oryzae* cakes from the *M. oryzae* plates. These *M. oryzae* cakes were smeared onto the medication-containing plates and cultured in a constant light incubator for 7 days at 28 °C with alternating light and dark conditions. The growth of *M. oryzae* was monitored and recorded throughout this period. Endophytic bacteria with strong activity levels were chosen as reference strains for further research based on the inhibition rate of each endophytic fermentation broth against *M. oryzae*, as determined by growth diameter. The inhibition rate (%) = [1−(diameter of *M. oryzae* in the treatment group-diameter of *M. oryzae* cake in the treatment group)/(diameter of *M. oryzae* in the control group-diameter of *M. oryzae* cake in the control group)] × 100. The strongest antifungal endophytic bacterium, designated B-612, was chosen for additional research.

### 4.4. Identification of Strain B-612

Standard procedures were used to recover genomic DNA from overnight B-612 cultures [37]. The gene encoding 16S rDNA can be used to distinguish bacterial species because it is the corresponding DNA sequence on the bacterial chromosome that encodes rRNA. 16S rDNA is extremely conserved in structure and function and is the most helpful and widely used molecular clock used in the systematic classification studies of bacteria [38]. Genomic DNA from strain B-612 was extracted using the sodium dodecyl sulfonate (SDS) technique for molecular characterization. We used forward primer 27F (5-AGAGTTTGATCCTGGCTCAG) and reverse primer 1492R(5-TACGGCTACCTTGTTACGACGACTT) to amplify the isolate’s 16S rDNA gene. The temperature protocol for the polymerase chain reaction (PCR) was 94 °C for 5 min, 94 °C for 30 s, 55 °C for 45 s, and 72 °C for 90 s for 30 cycles, with a final 10 min extension at 72 °C (Bio-Rad Powerpac300, Concord, CA, USA). Tsingke Biological Technology Corporation performed sequencing (Chengdu, China) and BLAST was used to analyze the 16S rDNA sequences (https://blast.ncbi.nlm.nih.gov, (accessed on 20 March 2023)). Then, we used MEGA 5.1 software (https://www.megasoftware.net/, (accessed on 20 March 2023)) to perform phylogenetic analysis of strain B-612.

### 4.5. Antibiotic Susceptibility Assay

The susceptibility of the isolate to probiotics was investigated by applying the procedure outlined by the Clinical and Laboratory Standards [39]. The fresh endophytic bacterial solution was evenly applied to the surface of a beef paste peptone plate; after the bacterial solution was dried, drug-sensitive tablets containing different antibiotics were taken and applied to the surface of the medium. The plates were incubated in a constant temperature incubator for 24 h at 28 °C to observe the sensitivity of bacteria to various antibiotics and to measure the diameter of the inhibition circle. The drug-sensitive tablets of the antibiotics used were purchased from Hangzhou Microbiological Reagent Co. The product name and the corresponding product item number are shown in Table 1.

### 4.6. Defense-Related Gene Expression

LTH plants were grown in a growth chamber (18 h of light at 28 °C and 6 h of darkness at 2 °C) to the three-leaf stage. Using distilled water with conidial suspension (concentration of 1 × 10^5^ conidia/mL) as the control group, B-612 fermentation solution and conidial suspension (concentration of 1 × 10^5^ conidia/mL) were mixed in a 1:1 ratio by volume and sprayed equally onto rice leaves. The rice leaves were collected at 0, 24, 48, and 72 h [40]. A Thermofisher NanoDROP was used to determine the quantity and quality of total RNA after it had been extracted using the Trizol technique. Then, we synthesized cDNA using a Primescript RT reagent kit from Takara (Beijing, China) in accordance with the manufacturer’s instructions. The qRT-PCRs were conducted using a BIO-RAD connect and *OsActin* expression levels were used as an internal standard for normalization. Table 2 shows the primer sequences used for key defense genes and the internal reference gene. SYBR Premix Ex TaqTM (TransGen Biotech, Beijing, China) was used for real-time PCR and each reaction was carried out three times.

### 4.7. H_2_O_2_ Accumulation

LTH rice leaves were collected at 0, 24, 48, and 72 h and stained for 12 h in the dark at pH = 3.8 with DAB, 1 mg/mL [41]. The stained leaves were then washed with distilled water after being destained with 95% ethanol until translucent. A ZEISS stereomicroscope was then used to track the accumulation of H_2_O_2_ in the leaves.

### 4.8. Detection of Lipopeptide Biosynthetic Genes

The template used for the detection of lipopeptide biosynthetic genes was endophytic genomic DNA that had been kept at −20 °C. We amplified surfactin, iturin, fengycin, and bacillomycin biosynthetic genes from genomic DNA by PCR. After being recovered, the target strips were delivered to DynaScience Biotech for sequencing evaluation. The primers used were described previously. The temperature protocol for the PCR was 94 °C for 5 min, 94 °C for 30 s, 55 °C for 45 s, and 72 °C for 90 s for 30 cycles, with a final 10 min extension at 72 °C. Table 3 shows the primer sequences used to identify the lipopeptide genes.

### 4.9. B-612 Fermentation Broth Crude Extraction

Under reduced pressure and at 40 °C, the culture filtrate (10 L) was concentrated to a dark-brown tarry residue. Thereafter, 30 L of each of the following were used to remove the dark-brown tarry residue: n-hexane, dichloromethane, ethyl acetate, and 1-butanol. Three extractions of each organic solvent were performed. Each organic extract was then processed through a rotating vacuum evaporator to create a paste, which was then dissolved in sterile distilled water to test for antifungal activity. The organic extract with the most potent antifungal properties was then chosen for the next stage of experimentation.

### 4.10. Germination Testing of M. oryzae Conidia and the Formation of Appressorium

The State Key Laboratory of Agricultural Gene Exploration and Utilization in Southwest China kindly contributed to *M. oryzae* Guy11. The conidia originated from Guy11, a 9-day-old plant raised on complete medium (CM). Using the 1-butanol crude extract of B-612 and distilled water, the spore concentration was adjusted to approximately 1 × 10^5^ conidia/mL [42]. Then, 50 μL of spore suspension was dropped onto the hydrophobic slide and kept at room temperature. With 100 conidia randomly chosen for observation, the germination rate of conidia and the development of appressorium were examined under a ZEISS fluorescence microscope at 2, 8, 12, 24, and 48 h. The experiment was repeated three times. The germination rate was calculated using the formula: Germination rate (%) = (A1/A2) × 100, where A1 represents the total number of conidia and A2 represents the number of conidia that had germinated. The developmental rate of appressorium was determined as follows: Appressorium formation rate (%) = (B1/B2) ×100 where B1 represents the number of conidia that formed appressorium and B2 represents the total number of conidia [43].

### 4.11. In Vivo Experiments with Living Leaves

An experimental paddy field was divided into many plots, each with an area of 1 m^2^. A film was employed to divide each plot to minimize the impact of various treatments. One-hundred LTH rice seeds were distributed uniformly throughout each plot before a 30-day cultivation period. Then, 150 mL of water, LB medium, carbendazim, B-612 bacterial solution, or B-612 filter fermentation solution, were sprayed into the plots. Each solution contained 0.1% of Tween 20 and each plot received 150 mL of *M. oryzae* conidia suspension containing 1 × 10^5^ conidia/mL, after one day.

### 4.12. Main Instrumentation

High temperature autoclaves(TOMY SS-325, Tokyo, Japan); Ultra Clean Bench(Airtech SW-CJ-1F, Zhejiang, China); Ultra-low temperature refrigerators(Thermo, Waltham, MA, USA); Constant Temperature Oscillating Incubator(Multitron Standard, Lausanne, Switzerland); Lighted incubators(ZQLY-180N, Shanghai, China); High-speed frozen centrifuge (Eppendorf 5145R, Hamburg, Germany); analytical balance (BS124S, Burkhardtsdorf, Germany); water purifier (Direct-Q3, Bay City, MI, USA); pH meter (PHS-3C, Nanjing, China); PCR instrument (Bio-Rad C1000, Hercules, CA, USA); electrophoresis instrument (Bio-Rad Powerpac 300, Hercules, CA, USA); gel imager (Bio-Rad ChemiDoc MP, Hercules, CA, USA); nucleic acid protein analyser (Beckman DU800, Missouri City, TX, USA); Zeiss stereo microscope (Car Zeiss Discovery.v20, Göttingen, Germany); qRT-PCR instrument (Bio-Rad connect, Hercules, CA, USA); rotary evaporator (Buchi. V-850, Allschwil, Switzerland))

The reagents used in this study were purchased from the Sichuan Ruijinte Technology company.

## Figures and Tables

**Figure 1 ijms-24-08513-f001:**
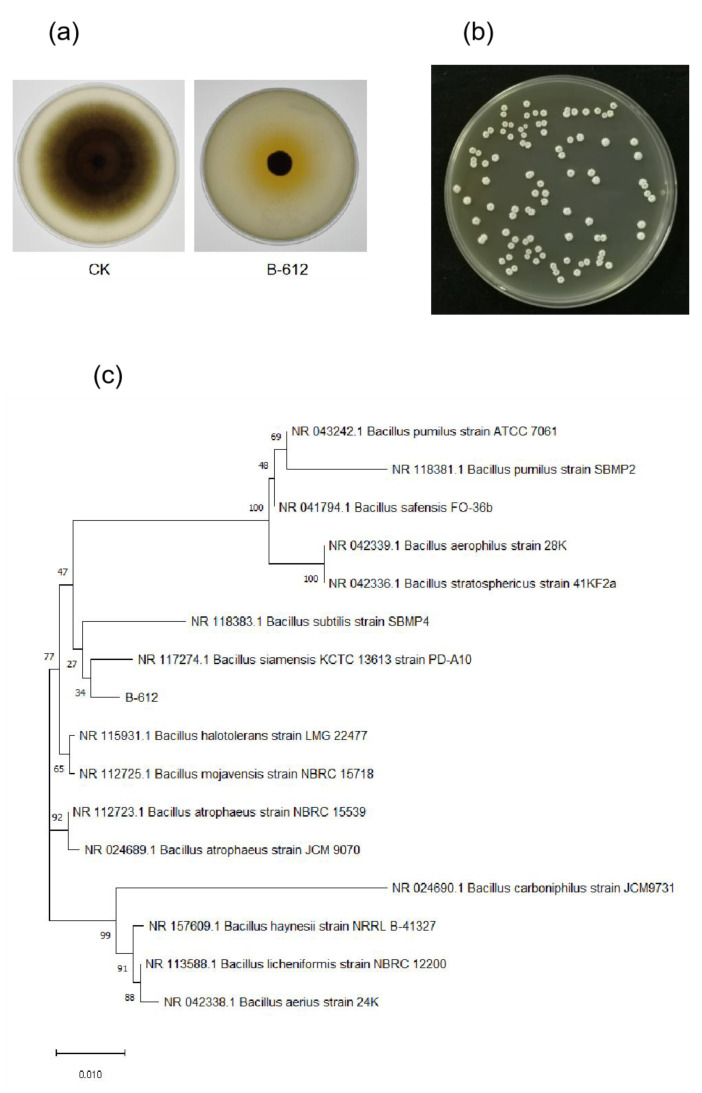
Isolation and identification of B-612. (**a**) B-612 showed a strong inhibitory effect against *M. oryzae* Guy11. (**b**) Colony morphology of strain B-612. (**c**) Phylogenetic tree of B-612. The phylogenetic tree was constructed by the neighbor-joining (NJ) method using MEGA 5.1 software.

**Figure 2 ijms-24-08513-f002:**
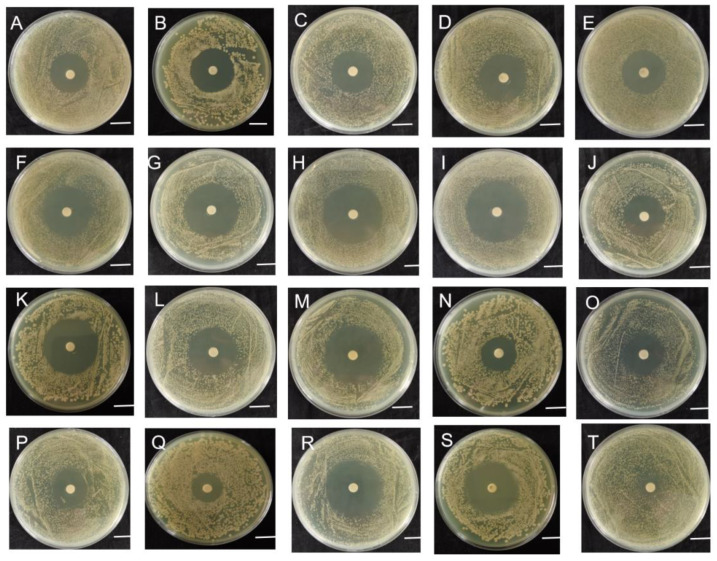
Sensitivity of strain B-612 to various antibiotics. A: penicillin; B: qxacillin; C: ampicillin; D: carbenicillin; E: piperacillin; F: cephalexin; G: cefamezin; H: cefradine; I: cefuroxim; J: ceftazidime; K: ceftriaxone; L: cefoperazone; M: doxycycline; N: amikacin; O: gentamicin; P: kanamycin; Q: neomycin; R: tetracycline; S: minocycline; T: erythromycin; scale, 10 mm.

**Figure 3 ijms-24-08513-f003:**
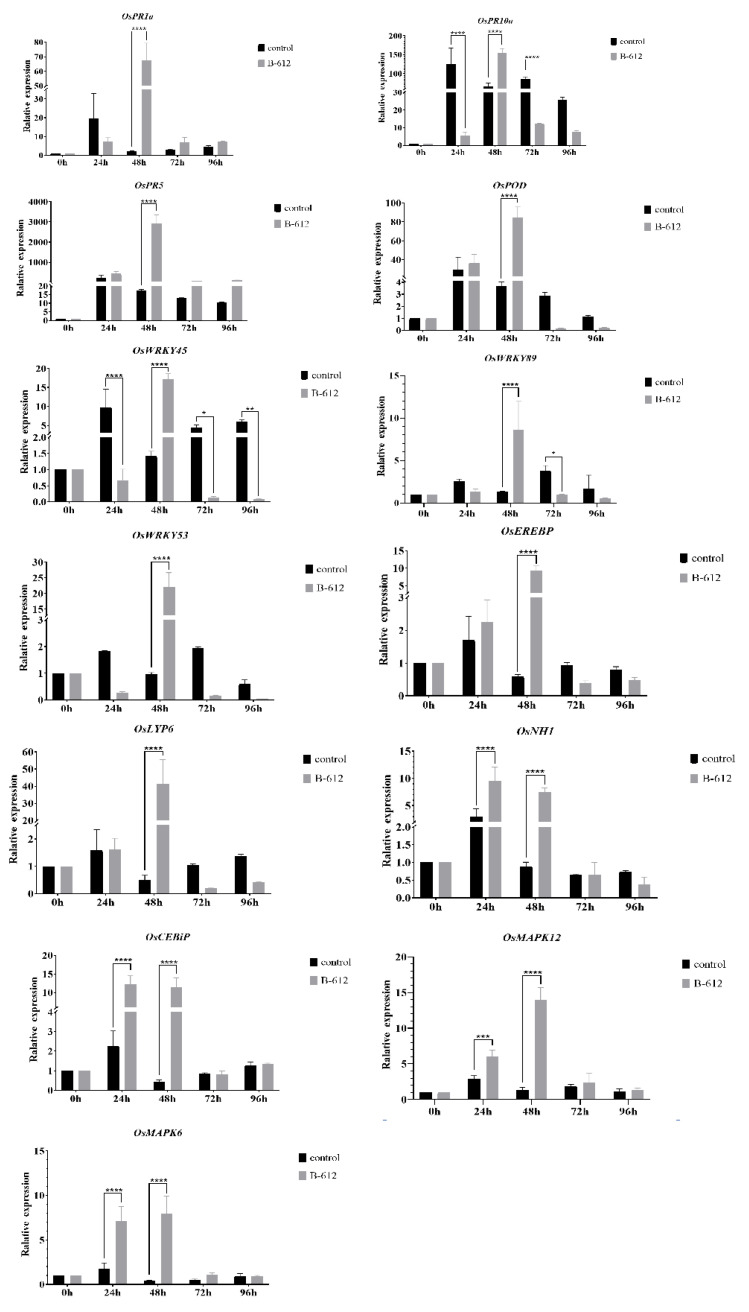
Effect of B-612 on the expression of genes related to rice defense. Multiple comparisons were used to analyze significant differences, *p* < 0.01. * difference is significant at the 0.05 level; ** difference is significant at the 0.01 level; *** difference is significant at the 0.001 level; **** difference is significant at the 0.0001 level.

**Figure 4 ijms-24-08513-f004:**
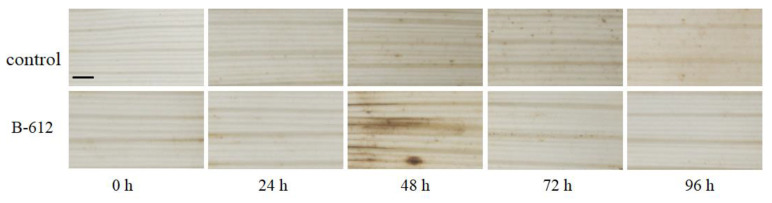
Differential content of H_2_O_2_ as indicated by diaminobenzidine (DAB) staining. Scale bar, 50 μm.

**Figure 5 ijms-24-08513-f005:**
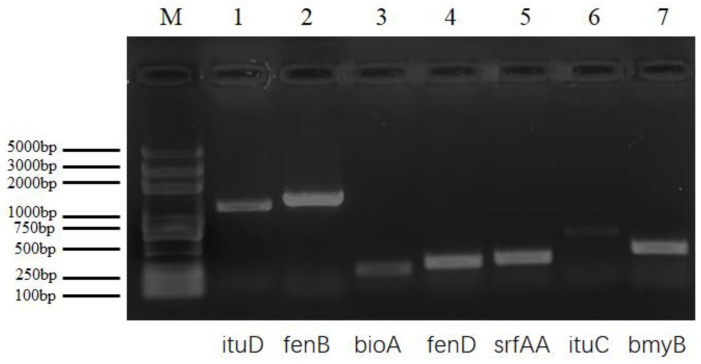
Electrophoresis and detection of genes related to antimicrobial lipopeptides.

**Figure 6 ijms-24-08513-f006:**
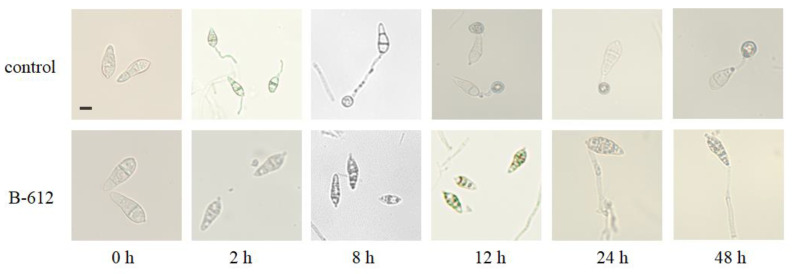
Effects of the antifungal substances of B-612 on *M. oryzae* conidia germination and the formation of appressorium. Morphological changes in the conidia of *M. oryzae* over different periods following treatment with 1-butanol crude extract of B-612. Scale bar, 20 μm.

**Figure 7 ijms-24-08513-f007:**
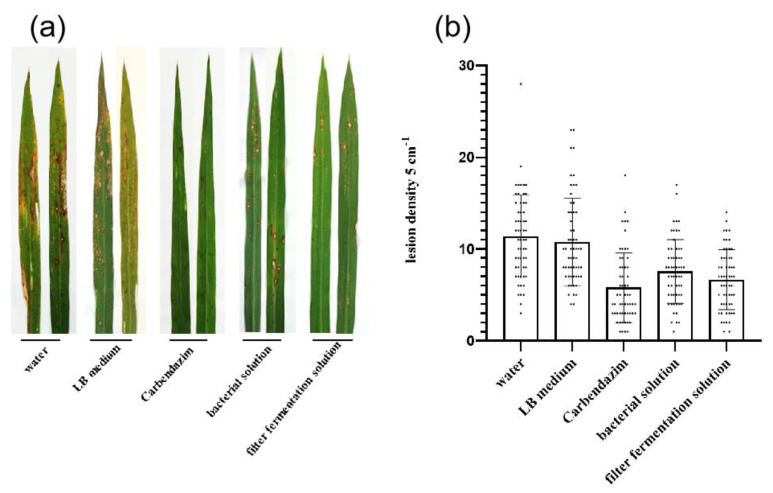
Evaluation of the protective effects of B-612 against rice blast under field conditions. (**a**) An outbreak of Xintuan leaf plague in Lijiang under different treatments. (**b**) The number of diseased spots within 5 cm of the leaf edge in Lijiangxintuan under different treatments; Statistically significant differences were analyzed by multiple comparisons (*p* < 0.01).

**Table 1 ijms-24-08513-t001:** The product name and the product number.

Product Name	Product Number
penicillin	S1001
qxacillin	S1002
ampicillin	S1003
carbenicillin	S1004
piperacillin	S1005
cephalexin	S1011
cefamezin	S1012
cefradine	S1013
cefuroxim	S1015
ceftazidime	S1019
ceftazidime	S1020
ceftriaxone	S1021
doxycycline	S1037
amikacin	S1027
gentamicin	S1028
kanamycin	S1030
neomycin	S1034
tetracycline	S1036
minocycline	S1038
erythromycin	S1039

**Table 2 ijms-24-08513-t002:** qRT-PCR primers.

Primer Name	Primer Sequences (5′—3′)
Actin	F:GAGTATGATGAGTCGGGTCCAG
	R:ACACCAACAATCCCAAACAGAG
PR1a	F:GCTACGTGTTTATGCATGTATGG
	R:TCGGATTTATTCTCACCAGCA
PR10a	F:AATGAGAGCCGCAGAAATGT
	RGGCACATAAACACAACCACAA
PR5	F:GGTACAACGTCGCCATGAGCT
	R:TGGGCAGAAGACGACTCGGTAG
CEBiP	F:CATCGCTCATCATACAAACCA
	R:GGAGATAACAGACATGCTCCAC
LYP6	F:TGCCCAGGACCACATCAGT
	R:CCAGGGAAGCCCGGAATAT
NH1	F:AAGCGGTTCAAATCTCAAA
	R:GCCTCCATCGGAAACATA
MAPK6	F:CTCGTACCACCTCAGAAAC
	R:AAATACAGCCCACAGACC
MAPK12	F:ATCGCTTCAAACGACAGT
	R:GTGACATTGGAGGGCTTA
POD	F:GGCCTTGGCAAATACCGACC
	R:TCGTGTGTGCTCCTGAGAGA
WRKY45	F:GCAGCAATCGTCCGGGAATT
	R:GCCTTTGGGTGCTTGGAGTTT
WRKY53	F:ACGGGCAGAAGCAGGTGAAG
	R:CCCTTGTAGACGATCTGGGTGA
WRKY89	F:GCACCTCACAATGATGGA
	R:GGACAGCCTTGCACTTTA
EREBP	F:GTGTTCGTGTCTGGCTTGG
	R:CACTTGACTTGGGTGCTTTA

**Table 3 ijms-24-08513-t003:** PCR primers of antifungal lipopeptide-related genes.

	Primer Sequences(5′—3′)
bioA	F:TTCCACGGCCATTCCTATAC
	R:TTTGTCCCCTTATCCTGCAC
SrfA	F:GAAAGAGCGGCTGCTGAAAC
	R:CCCAATATTGCCGCAATGAC
FenD	F:CCTGCAGAAGGAGAAGTGAAG
	R:TGCTCATCGTCTTCCGTTTC
FenB	F:CTATAGTTTGTTGACGGCTC
	R:CAGCACTGGTTCTTGTCGCA
ItuC	F:TTCACTTTTGATCTGGCGAT
	R:CGTCCGGTACATTTTCAC
ItuD	F:ATGAACAATCTTGCCTTTTTA
	R:TTATTTTAAAATCCGCAATT
bmyB	F:TGAAACAAAGGCATATGCTC
	R:AAAAATGCATCTGCCGTTCC

## Data Availability

No new data created.

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
