# Peer review of "Isolation of Bacillus siamensis B-612, a Strain That Is Resistant to Rice Blast Disease and an Investigation of the Mechanisms Responsible for Suppressing Rice Blast Fungus"

_ijms, 2023, doi:10.3390/ijms24108513_

Round 1
Reviewer 1 Report
In this work, authors isolate a previously uncharacterized endophytic strain of Bacillus siamensis that they name B-612. Through 16s rRNA analysis, authors confirm this strain as a Bacillus siamensis and they furthermore show that B-612 is almost identical to KCTC13613.
Authors further characterize B-612 by testing its antibiotic susceptibility, plant defense promoting effect by expression level of known defense genes by RT-qPCR, H2O2 accumulation, and assessment of presence for known bio-controlling lipopeptide biosynthetic genes.
Finally, by using B-612 inoculation or crude extract, they show a growth inhibition effect of B-612 crude extract on Magnaporthe oryzae conidia and appressorium formation, as well as an increase in tolerance to M. oryzae on rice plants pre-inoculated with B-612.
The work is well-written, clearly presented, and experiments show unambiguous results.
Some issues I would like addressed:
First of all, I would suggest to authors to provide a more detail description of statistical method used (mainly, the test used on each statistical analysis, number of data points analyzed, and statistical design)
Authors show that B-612 is susceptible to a range of antibiotics:
B-lacmates (Penicillin – ampicillin, benzoxicillin, piperacillin), cephalosporins (cephalexin, cefazolin, cefradine, cefuroxime, ceftazidime, ceftriaxone, cefoperazone), aminoglycosides (gentamicin, kanamycin, neomycin), and tetracyclines (tetracycline, minocycline). Which lead to the conclusion that this susceptibility implies that is less likely to become drug-resistant bacteria. I do not see enough evidence for this claim, as the authors do not provide genetic, gnomic, or evolutionary experiments (such as the likely of DNA transfer events with other species to happen, a full genome sequence to show absence of inactive resistance genes that could get activated, or forced evolution assays under antibiotic pressure. However strong and unsupported as a claim, it does not affect the overall research. Though, authors could further increase the list of antibiotics to include the full spectrum of chemical families since they only tested b-lactamates, cephalosporins, aminoglycosides, and tetracyclines, lacking sulfonamides, quinolones, among others.
Moreover, I could not find info on: carbobenzillin, polytetracycline, butamicarna. Therefore, it would be of great use for other researchers if the authors provide the manufacturer and catalog number of all the assayed antibiotics.
Authors show a great induction of defense genes upon inoculation of rice with B-612. This could be made even more informative by comparing the extent of this activation by assessing the expression of same genes upon inoculation with a non-pathogenic, non-defense inducer bacterium, and also comparing to a known PGP bacillus (such as B. siamensis KCTC13613).
Authors claim that “B-612 expressed many relevant and functional genes, including fengycin (fenB, fenD), iturin (ituD), surfactin (srfAA), bacillomycin 142 (bmyB), and biotene (bioA).” However, I failed to find the experiment that confirms induction on expression levels for these genes (i.e., RT-qPCR or alike). Moreover, authors claim that “B-612 is able to synthesize fengycin synthetase B and fungogenin synth”. But no experiment that shows the synthesis of these products is provided. Same for iturin production.
For the B-612 antifungal activity on M. oryzae assay, authors use water as control, but the B-612 extract is made using 1-butanol. Therefore, control assay should also use 1-butanol to assess the effect of extraction media on the fungus.
An overall suggestion for future work is to sequence B-612 genome to confirm that is not a previously characterized B. siamensis strain.
Minor comments:
Line 77: missing a space between “Guy11” and “The”
Line 207-208: Authors describe that “Xu et al. purified iturin A and bacillomycin F from Bacillus siamensis JFL15 to inhibit the growth of M. oryzae, but did not investigate the precise mechanism involved”. However, I don’t see the reason to bring this claim as the author’s work does not investigate the precise mechanism involved in B-612 inhibition of M. oryzae growth either.
Reviewer 2 Report
I checked your manuscript and described comment below.
Rice blast is an important disease that causes great damage to rice, which is an agricultural product.
This paper is very meaningful because it summarizes the papers for acquiring resistance to blast disease by infecting Oryza sativa L with Bacillus siamensis B-612.
The author uses MEGA5.1 to analyze Figure1. The current version of MEGA is MEGA11. If it is a normal research paper, it is better to re-analyze it, but this paper is an essay, I think there is no problem.
I don't think this paper has any major mistakes or grammatical problems.
Reviewer 3 Report
In the manuscript entitled “Isolation of Bacillus siamensis B-612, a strain that is resistant to rice blast disease and an investigation of the mechanisms responsible for suppressing rice blast fungus”, the authors showed that a Bacillus siamensis strain B-612 isolated from cauliflower leaves is inhibitory to Magnaporthe oryzae growth, a filamentous ascomycete fungus which is a devastating pathogen to rice crop. While rice is one of the most economically important crops globally, it is often susceptible to devastating yield loss due to Magnaporthe oryzae, commonly referred to as rice blast. Control methods for rice blast include chemical, biological, and breeding for resistant rice varieties. However, there are some drawbacks to resistant varieties since the resistance can break down over time, while chemical control methods can incur high production costs and can also be detrimental to the environment. Thus, biological alternatives provide a more environmentally friendly way for controlling this pathogen. In this study, between 24 and 48 hours after inoculation with B-612 fermentation solution, expression of defense related genes were significantly upregulated and peroxidase activity increased by 80-fold in 48 hours. This result demonstrated that the rice defense mechanism critical for protection against the disease is activated in response to B-612 exposure. Furthermore, in field trials, when seedlings were treatment with B-612 fermentation solution or B-612 bacterial solution prior to infection with the fungus, the severity of the disease was significantly inhibited. Overall, this study indicates that Bacillus siamensis B-612 is a potential biological control agent which could be effective in preventing blast disease if rice plants are treated during the pre-susceptible stage.
The main question addressed by this study was to determine the degree to which the endophytic bacterium Bacillus siamensis B-612 inhibits the growth of the rice pathogen Magnaporthe oryzae (rice blast) as well as to elucidate the mechanisms for this growth inhibition.
This study is relevant to the field since rice crop is of global economic importance and producers suffer significant economic loss due to rice blast disease which often severely impacts yield production. Currently, chemical and biological agents exist to control this pathogen along with resistant rice varieties. However, resistant varieties can break down over time because of rapid development of new strains of the pathogen, while chemicals can incur high production costs and have detrimental environmental consequences. Thus, there is always a need for additional effective biological agents that are simple to grow and manipulate to protect against this pathogen.
The results presented in this study suggest for the first time that Bacillus siamensis B-612 is a strong potential biocontrol agent against rice blast. This was demonstrated in field trials where, after treatment with B-612 fermentation solution, leaves were less susceptible to blast. The study also provides new insights into the mechanisms by which B-612 inhibits blast growth. qRT-PCR results reveal significant upregulation in expression for genes associated with rice defense mechanisms within 24 – 48 hours after inoculation with B-612.
The authors should consider adding more information to the methodology regarding the field trials. How many times were the field trials conducted? How many different time points during seedling development were the leaves analyzed for blast lesions? Were the plants analyzed for yield?
No other controls are needed for the scope of this study. The authors used appropriate controls in all of their experiments.
The conclusions are consistent with the results provided. All the experiments conducted were relevant to the main question the authors were addressing.
The references are appropriately relevant to the study.
Some of the labels for the figures should be edited to improve clarity.
Also, should “carbobenzillin” be “carbenicillin”?
Overall, the authors’ results support their conclusions, and the analyses of the data are adequate for the overall objective of the study.
